# Individualization of Intensity Thresholds on External Workload Demands in Women’s Basketball by K-Means Clustering: Differences Based on the Competitive Level

**DOI:** 10.3390/s22010324

**Published:** 2022-01-01

**Authors:** Sergio J. Ibáñez, Carlos D. Gómez-Carmona, David Mancha-Triguero

**Affiliations:** 1Research Group in Optimization of Training and Sports Performance (GOERD), Didactics of Body Expression, Music and Plastic Department, Sport Science Faculty, University of Extremadura, 10005 Caceres, Spain; sibanez@unex.es; 2Section of Sport Technical Formation, Center of Professional Training CESUR, 30007 Murcia, Spain; 3Department of Sport, CEU San Pablo University Cardenal Spínola, 41930 Bormujos, Spain

**Keywords:** team sports, technology, speed, impacts, k-means cluster

## Abstract

In previous studies found in the literature speed (SP), acceleration (ACC), deceleration (DEC), and impact (IMP) zones have been created according to arbitrary thresholds without considering the specific workload profile of the players (e.g., sex, competitive level, sport discipline). The use of statistical methods based on raw data could be considered as an alternative to be able to individualize these thresholds. The study purposes were to: (a) individualize SP, ACC, DEC, and IMP zones in two female professional basketball teams; (b) characterize the external workload profile of 5 vs. 5 during training sessions; and (c) compare the external workload according to the competitive level (first vs. second division). Two basketball teams were recorded during a 15-day preseason microcycle using inertial devices with ultra-wideband indoor tracking technology and microsensors. The zones of external workload variables (speed, acceleration, deceleration, impacts) were categorized through k-means clusters. Competitive level differences were analyzed with Mann–Whitney’s U test and with Cohen’s d effect size. Five zones were categorized in speed (<2.31, 2.31–5.33, 5.34–9.32, 9.33–13.12, 13.13–17.08 km/h), acceleration (<0.50, 0.50–1.60, 1.61–2.87, 2.88–4.25, 4.26–6.71 m/s^2^), deceleration (<0.37, 0.37–1.13, 1.14–2.07, 2.08–3.23, 3.24–4.77 m/s^2^), and impacts (<1, 1–2.99, 3–4.99, 5–6.99, 7–10 g). The women’s basketball players covered 60–51 m/min, performed 27–25 ACC-DEC/min, and experienced 134–120 IMP/min. Differences were found between the first and second division teams, with higher values in SP, ACC, DEC, and IMP in the first division team (*p* < 0.03; *d =* 0.21–0.56). In conclusion, k-means clustering can be considered as an optimal tool to categorize intensity zones in team sports. The individualization of external workload demands according to the competitive level is fundamental for designing training plans that optimize sports performance and reduce injury risk in sport.

## 1. Introduction

The training process in team sports is based on a rigorous and methodical process for designing the external and internal workload administered to the players [1]. The management of training workload is essential to optimize sports performance, reduce injury risk, and avoid overtraining, as well as to monitor the evolution of the players’ physical level throughout their sport career [2]. For this purpose, the recording of internal and external workload is fundamental for obtaining objective data on the training process in team sports, and specifically in basketball [3].

Internal and external workload in basketball can be quantified using different methods depending on the available resources [4]. For time-motion analysis, the first method utilized was video-tracking [5], but due to the high cost and difficult data processing, this method has been largely replaced by radiofrequency technologies in indoor conditions with antennas as reference system such as ultra-wideband (UWB) or Bluetooth [6]. These data can be complemented with microsensors to record how the players’ movements influence the workload supported by the musculoskeletal structures of the body [7]. Regarding internal workload, the most extended method is heart rate telemetry, although different hematological markers such as blood lactate, cortisol, insulin, and glucose have also been measured [4].

The monitoring of workload will allow team staff to ascertain the mechanical and locomotor stress suffered during efforts as well as the biological reaction of the player’s body [8]. Following the principles of training, the dose-response is individually based on physiological (e.g., age, physical fitness), psychological (e.g., perceived effort, motivation), environmental (e.g., sport discipline, competitive level, period of the season), and genetic (e.g., sex) factors [9]. Thus, the thresholds of intensity zones related to speed, acceleration, deceleration, and impacts suffered by the players should be adapted to the individual characteristics of the athletes [10].

Previous research in basketball identified five speed zones ([1] standing/walking, 0–6 km/h; [2] jogging, 6.1–12.0 km/h; [3] running, 12.1–18.0 km/h; [4] high-speed running, 18.1–24.0 km/h; [5] sprinting, >24.1 km/h) [11]; two ([1] low, <3 m/s^2^; [2] high, >3 m/s^2^) or three ([1] low, 1–2.5 m/s^2^; [2] moderate, 2.5–4 m/s^2^; [3] high, >4 m/s^2^) acceleration/deceleration zones [12,13]; and (1) low intensity (<3 m/s^2^) and (2) high intensity (>3 m/s^2^); or five impact zones ([1] very low, 0–3 g; [2] low, 3–5 g; [3] moderate, 5–8 g; [4] high, 8–10 g; [5] very high, >10 g) [14]. The main problem is that the intensity thresholds of sport locomotion have been defined by the companies that develop sport technology in their own software [15], adapted from previous studies [16] or determined by authors’ criteria [17], making necessary threshold individualization considering the sport discipline, and the players’ level or sex [10].

For this purpose, recent studies proposed different methods to calculate intensity thresholds such as: (a) based on maximum values reported in the literature combined with own competition and training data [18], (b) using Gaussian distributions with unknown parameters [19], (c) applying the k-means clustering algorithm [10], or (d) through a spectral clustering algorithm [20]. In basketball, only one previous study has used a k-means clustering algorithm to individualize speed zones in women youth players (standing, <3.6 km/h, walking: 3.6–6.5 km/h, jogging: 6.5–10.2 km/h, running: 10.2–14.4 km/h, sprinting: >14.4 km/h) [21], but there is no approximation to the individualization of intensity thresholds in accelerations, decelerations, and impacts.

Therefore, due to the importance of individualized thresholds to identify the specific values based on sport discipline, players’ level, or sex [22], as well as the lack of research about the individualization of speed, acceleration, deceleration and impact zones in basketball, and specifically in women’s basketball, the purposes of the present study were: (a) to conduct an individualization of the work zones of each variable in two professional women’s basketball teams, using the k-means cluster algorithm to ascertain the different training intensities in SP, ACC, DEC and IMP; (b) to characterize the external workload demands of the selected variables related to the 5 vs. 5 profile in the total and normalized variables, and (c) to compare the external workload in a 5 vs. 5 game situation in the training sessions according to the competitive level (first vs. second division).

## 2. Materials and Methods

### 2.1. Design

This research is classified within the empirical studies that follow an associative strategy through a cross-sectional comparative design [23] that explore the thresholds of intensity zones in external workload, characterize the performance of women’s basketball players in training games (5 vs. 5) as well as examine the differences between teams according to the competitive level (first vs. second division).

### 2.2. Participants

Twenty-two professional women’s basketball players that belonged to two elite-level teams (first division, Liga Femenina 1; second division, Liga Femenina 2) participated in the present study (first division, *n* = 10, age = 22.51 ± 2.68 years, height = 1.81 ± 0.08 m, body mass = 75.58 ± 12.32 kg; second division, *n* = 12, age = 21.79 ± 2.45, height = 1.77 ± 0.09 m., body mass = 78.32 ± 11.55 kg) They were evaluated during 15 days of the preseason period in which they performed 10 training sessions. The players that took part in the present study met the following inclusion criteria: (a) they had participated in all training sessions during the 15-days microcycle and in all tasks in each session; (b) they had not presented musculoskeletal injuries and health problems in the previous two months; (c) they were familiarized with high-level monitoring during more than 10 training sessions or competitive games; and (d) they had played at the maximum competitive level in any country for at least 2 years before the study.

The players, coaches and managers of the teams were informed before the investigation about the possible risks and benefits of participation. An informed consent form was signed by the coaching staff, managers, and basketball players of each team. The research was carried out under the criteria of the Declaration of Helsinki (2013) and was approved by the Bioethics committee of the University (233/2019).

### 2.3. Variables

For this research, the competitive level of the teams (first women’s division vs. second women’s division) was considered as an independent variable. For the evaluation and to achieve the proposed objectives, the following dependent variables were chosen that are widely used in basketball [11,12,24]:*Distance covered:* Total number of meters covered by the player classified according to the speed of locomotion.*Number of accelerations*: Total number of positive increases in speed that each player performed (measured in m/s^2^), classified according to intensity.*Number of decelerations*: Total number of negative increases in speed that each player performed (measured in m/s^2^), classified according to intensity.*Number of impacts*: Total number of impacts received by each player (measured in g forces), classified according to the magnitude of g force supported.

All the variables were grouped into five work zones by the k-means clustering algorithm.

### 2.4. Equipment

Data were recorded with WIMU PRO^TM^ inertial devices (RealTrack Systems, Almeria, Spain). Distance covered at different speeds, accelerations and decelerations were obtained through ultrawide-band (UWB) tracking technology at 33 Hz. This UWB system was designed to replace the satellite reference in indoor conditions [25] and consisted of a transmitter reference system (antennas) and receiver (devices). The reference system is composed of eight antennas placed in the corners (*n =* 4, 5 m from the perimeter), on the middle line (*n* = 2, 7 m from the perimeter) and behind the baskets (*n* = 2, 7 m from the perimeter), forming an octagon and positioned at a height of 3 m. Switch-on and calibration processes were performed following the manufacturer’s recommendations that presented almost perfect validity and reliability [26]. Regarding impacts, the inertial device was composed of different microsensors (four accelerometers: 2× ± 16 g, 1× ± 32 g and 1× ± 400 g; three gyroscopes 2000°/s; one magnetometer) that were set at 100 Hz and presented almost perfect validity in accelerometer raw data [27]. Devices were located at the inter-scapular level in each player with an anatomical harness. The registered data were analyzed through the SPRO^TM^ software (RealTrack Systems, Almeria, Spain).

### 2.5. Procedures

First, the clubs were contacted to inform them about the study purposes and to invite them to participate. Once the proposal was accepted, an informed consent form was signed by coaches and players. The teams performed five training sessions on the court and one competitive game each week. During training sessions, the women’s basketball players were monitored with the inertial devices. Firstly, the antennae system was installed around the court and then the devices were placed 30 min prior to the session in a neoprene anatomical vest at the scapular level. During sessions, S VIVO^TM^ specialized software was used for the time selection of each task. After the end of each session: (1) data were downloaded to a laptop, (2) data were introduced into SPRO^TM^ manufacturer’s software to export external workload variables, (3) external workload variables were uploaded to the WIMU cloud storage and, (4) the report of the session was generated, and an informative dossier was sent daily for the team staff detailing the relevant information of the session. When the analysis and team report were completed, all the tasks performed by both teams involving five vs. five game situations on the full court of play were selected. Although the monitoring was during all training sessions, in this research only the five vs. five situations were analyzed. The duration of the task could vary depending on different aspects (first division team ≈ 8 min, second division team ≈ 6 min). However, when taking a break (to rest or perform another task, for example free throw shots), the task ended and if it was carried out again after the break, it was re-analyzed as another task (several data collections could be obtained in the same training session).

The raw data from each of the training tasks was analysed using the k-means clustering algorithm. The results of this algorithm were used to configure the SPROTM software with the specific ranges in which the external load variables of the research were classified. 

Finally, the results were obtained for each of the research variables classified into the five groups defined in the k-means cluster specific to the population of professional women’s basketball players, first division, second division and combined. These results have served to identify the differences in the external load in players at two different competitive levels.

### 2.6. Statistical Analysis

Firstly, data raw of total acceleration (AcelT, sum vector of acceleration in the three planes of movement, g force) and UWB speed (km/h) channels, as well as acceleration and deceleration values (m/s^2^) of each positive and negative change of direction generated by all players during sessions were imported to the statistical package. Three analyses with the k-means clustering algorithm based on five zones following previous basketball research were conducted [21]: (1) first division team, (2) second division team, and (3) total team data. In addition, the results pertaining to the quality of the k-means clustering are shown

Then, distance covered in each speed zone, number of accelerations and decelerations in each speed zone and number of impacts at each intensity were exported as total (accumulated value in each task) and relative (accumulated value in each task divided by the total time in minutes) variables to characterize the volume and intensity of tasks, respectively. Data normality and homoscedasticity were explored with the Kolmogorov–Smirnoff and Levene tests, showing a non-parametrical distribution. For this reason, external workload variables were characterized in the descriptive analysis as median and range (upper and lower values) and in plots as a histogram. 

Finally, the Mann–Whitney U test was conducted to analyze the effect of competitive level in total and relative variables. The effect size of differences was obtained with Cohen’s d and interpreted as follows: *d* < 0.2 as *trivial*, *d* = 0.2–0.5 as *small*, *d* = 0.5–0.8 as *moderate*, and *d* > 0.8 as *large* [28]. Statistical differences were considered if *p* < 0.05. Data analysis was performed using the Statistical Package for the Social Sciences (SPSS, IBM, SPSS Statistics, v.25.0 Armonk, NY, USA) and graphs were made using Prism software (GraphPad Software, San Diego, CA, USA).

## 3. Results

Table 1 presents the results of the k-means cluster analysis in five groups, following previous studies that have been referenced and that have been used in this process.

Table 2 shows the results pertaining to the quality analysis of the k-means clustering carried out in the research.

Figure 1 and Figure 2 show the histograms on the variables related to distance and impacts, and accelerations and decelerations, respectively. The results are similar in both teams analyzed, although the first division team presents a greater number of high intensity actions than the second division team.

Table 3 shows the external workload variables according to the volume of demands in the women’s basketball teams. Significant differences were observed according to competitive level (first and second division) in all variables analyzed except the highest intensity zone in accelerations and decelerations (*p* > 0.31). The effect size obtained is also high in all the variables except for the variables that do not show significant differences.

Table 4 shows the results of the external load variables according to the intensity of the demands. Statistical differences were found between competitive level in distance covered at jogging and running, in accelerations at total, low and moderate intensity, in decelerations at total, low, moderate and high intensity, and in impacts at all intensities with higher values in the first division team. Furthermore, the effect size shows high values in decelerations (*d* = 0.57–0.84) and low values in the rest of variables (*d* < 0.46).

## 4. Discussion

The control and quantification of the loads that the player supports during training or competition is a topic on the rise in recent years. However, this object of study is notably reduced when the selected sport is basketball and the sample comprises women players [24]. For this, the use of technology such as inertial devices has been facilitated and linked to the scientific and professional field of training in search of common lines of investigation [7]. However, in most cases, variables that are not optimal for the selected population are used to quantify the demands that an athlete supports [29,30]. Part of this problem is the generic use of ranges of these variables (volume or intensity) without individualizing the sample [31], causing serious consequences that affect the planning and results of a team during the competition. Therefore, the use of k-means clustering will help the results of the analyzed players to be individualized and established optimally as previous research existing in the literature [21,32], following one of the main principles of training.

Reviewing the literature, there are different investigations that use cluster analysis [11]. However, it is not a common practice due to the need for the previous data that are required for the analysis [21,32]. According to the results, differences were obtained in this research because of the individualization of the competitive process (sex, competitive level or characteristics of the sample) [11]. All aspects have an impact on the demands that players support, and therefore, on the workload demands performed during the game [12]. In contrast to using k-means clusters, the vast majority of investigations adjust their ranges of variables according to other investigations (without considering the individual characteristics of the sample) or with the values provided by the manufacturer [29]. This is an error that eliminates the individualization of the training process, which is one of the main objectives of the quantification and control of the competition and training workloads.

Regarding the demands that the analyzed players supported, differences are observed in the recorded variables depending on the competitive level, both in their volume and intensity. The studies that use k-means clustering to individualize speed thresholds have been carried out in different contexts (formative players, PE lessons, youth national-level women players) from the present study (five vs. five in full-court without breaks). González-Espinosa et al. [32] evaluated under-12 basketball players during four vs. four games in school competition, finding four work zones (<6 km/h; 6–12 km/h; 12–18 km/h; >18 km/h). Similarly, Gamero et al. [33] established four work zones through cluster analysis as follows (<5.2 km/h; 5.2–10.5 km/h; 10.5–15.7 km/h; >15.7 km/h). Along the same lines, García-Ceberino et al. [34] employed the K-means clustering method with five levels to differentiate PL/min load ranges in the context of PE, as load ranges should be adapted to the study population. Finally, Reina et al. [21] performed a cluster analysis during a competition in women players that were in their last formative stage (youth, close to amateur age) and the values were grouped into five categories (<3.6 km/h; 3.6–6.5 km/h; 6.5–10.2 km/h; 10.2–14.4 km/h; >14.4 km/h). The differences between previous research and the present study could be related to the multitude of cases collected in different training sessions in the same sample, as well as the specific characteristics of the players and playing context (e.g., minutes played, competitive level of the rival, players’ ages, players’ sex). These differences confirm the importance of individualization of the process and variability depending on the selected sample.

Finally, the main strength of this research was that it is the first approach of the utilization of k-means clustering to obtain reference values and individualize the training processes in first and second division women’s basketball players in five vs. five training tasks during a competitive period. In addition, they provide information on the workload that the players supported in the face of the same stimulus (training game) in which the demands are different depending on the competitive level. Therefore, this research provides relevant and specific information for the basketball team staff and sport scientists that may be a booming topic in coming years due to the exponential growth of Big Data and the continuous search for the individualization of a workload based on players’ characteristics and contextual aspects of the analyzed teams. The initial hypothesis on the research is that the five vs. five situations during training try to resemble competition situations, although this is not always achieved. Therefore, players with a good sport level, first or second division, have similar responses during training, identifying significant differences in their behaviors, although these are not very large.

## 5. Conclusions

From the results obtained in the present study that carried out a first approach to individualize work zones in first and second division women’s basketball players through a k-means cluster algorithm, different conclusions and practical applications can be drawn:Five work zones in speed (<2.31, 2.31–5.33, 5.34–9.32, 9.33–13.12, 13.13–17.08 km/h), acceleration (<0.50, 0.50–1.60, 1.61–2.87, 2.88–4.25, 4.26–6.71 m/s^2^), deceleration (<0.37, 0.37 −1.13, 1.14–2.07, 2.08–3.23, 3.24–4.77 m/s^2^), and impacts (<1, 1–2.99, 3–4.99, 5–6.99, 7–10 g) were established through k-means clustering, performing the analysis without attending to the competitive level. Regarding the difference in results depending on the competitive level, we found that the first division team presented the following values in speed (<2.14, 2.14–4.93, 4.94–8.71, 8.72–12.55, 12.56–16.54 km/h), acceleration (<0.58, 0.58–1.86, 1.87–3.21, 3.22–4.58, 4.59–7.01 m/s^2^), deceleration (<0.62, 0.62–2.00, 2.01–3.42, 3.43–4.86, 4.87–6.51 m/s^2^), and impacts (<1, 1–1.99, 2–2.99, 3–3.99, 5–8 g), while in the second division team, the speed values were grouped in (<2.40, 2.41–5.90, 5.91–10.47, 10.48–14.99, >14.99 km/h), acceleration (<2.35, 2.35–5.16, 5.17–8.99, 9.00–12.76, 12.77–16.71 km/h), acceleration (<0.44, 0.44–1.44, 1.45–2.64, 2.65–3.65, 3.66–4.61 m/s^2^), deceleration (<0.52, 0.52–1.74, 1.75–3.01, 3.02–4.22, 4.23–5.65 m/s^2^), and impacts (<1, 1–1.99, 2–2.99, 3–4.99, 5–8 g). The characterization of women’s basketball demands is important to quantify specific training and competition workloads adapted to the physical fitness of players and sport discipline. The use of mathematical processes like k-means clustering helps the establishment of thresholds in an objective way.Differences in speed, changes of speed and impact thresholds were found between categories. First division players obtained higher thresholds in all types of movements and at all intensities, especially in changes of speed. Therefore, the competitive level is one of the aspects to consider when work zones are individualized.The competitive level also affected the duration of five vs. five full-court training tasks. The first division team performed five vs. five full-court tasks with longer duration (~8 min, (7′30″ to 8′40″) than the second division team (~6 min). For this reason, a higher volume of distance, accelerations, decelerations and impacts at all intensities were found in the first division team. The adaptation of task duration is fundamental for achieving the desired training objectives. A lower competitive level is normally linked to lower physical fitness so that shorter tasks are necessary to maintain high intensity with longer between-repetition breaks.When variables were relativized to task duration, the first division team players covered a greater distance jogging and running, recorded higher positive and negative changes of speed and impacts at all intensities except in the very high zone. In this respect, the competitive level not only affected the workload volume in relation to task duration, but the movements were performed at higher intensity in the first division players. Therefore, an adaptation of the training, both in volume and intensity, is necessary to achieve the desired performance enhancement and reduce injury risk.

The main limitations of the research focus on the analysis of two teams in five vs. five situations in training sessions and not during competition. In addition, the analysis was carried out during the pre-season period. In order to improve the results obtained, it would be interesting for future research to make a comparison between five vs. five in competition and training sessions, as well as at different times of the season.

## Figures and Tables

**Figure 1 sensors-22-00324-f001:**
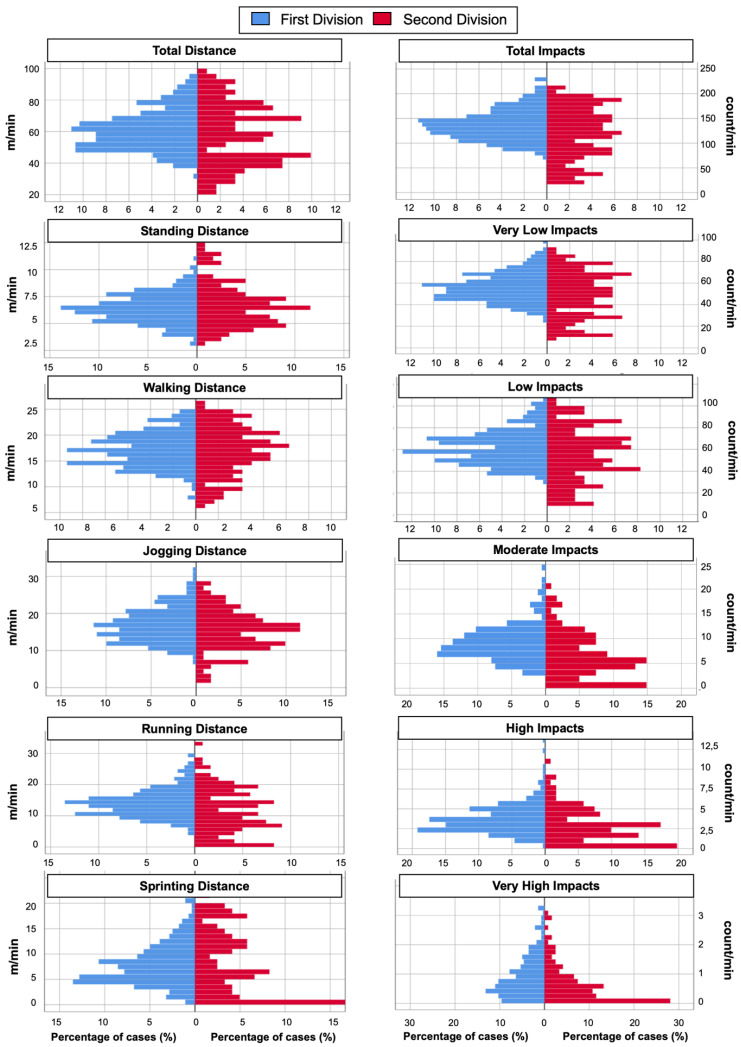
Histogram to represent the external workload demands in five vs. five in women’s basketball in the different speed and impact zones.

**Figure 2 sensors-22-00324-f002:**
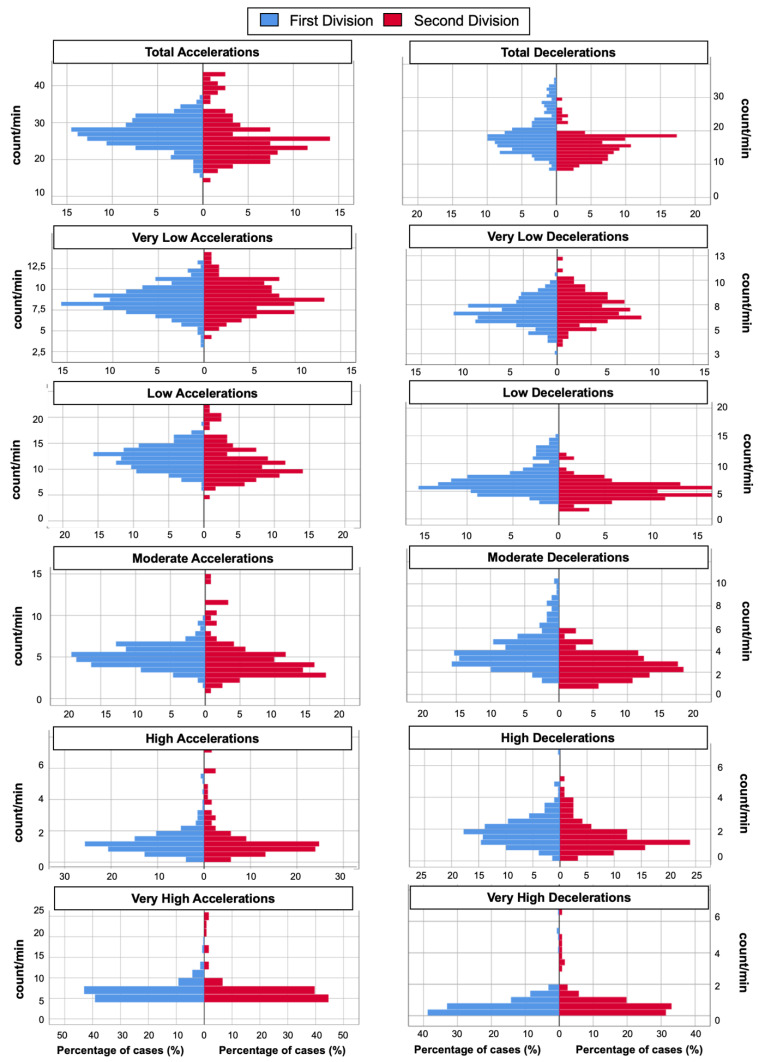
Histogram to represent the external workload demands in five vs. five in women’s basketball in the different acceleration and deceleration zones.

**Table 1 sensors-22-00324-t001:** Thresholds of external workload variables in female basketball according to competitive level.

Variables	Players’ Level	Very Low/Standing	Low/Walking	Moderate/Jogging	High/Running	Very High/Sprinting
Speed (km/h)	First Division	<2.14	2.14 to 4.93	4.94 to 8.71	8.72 to 12.55	12.56 to 16.54
	Second Division	<2.35	2.35 to 5.16	5.17 to 8.99	9.00 to 12.76	12.77 to 16.71
	Combinate	<2.31	2.31 to 5.33	5.34 to 9.32	9.33 to 13.12	13.13 to 17.08
Accelerations (m/s^2^)	First Division	<0.58	0.58 to 1.86	1.87 to 3.21	3.22 to 4.58	4.59 to 7.01
	Second Division	<0.44	0.44 to 1.44	1.45 to 2.64	2.65 to 3.65	3.66 to 4.61
	Combinate	<0.50	0.50 to 1.60	1.61 to 2.87	2.88 to 4.25	4.26 to 6.71
Decelerations (m/s^2^)	First Division	>−0.62	−0.62 to −2.00	−2.01 to −3.42	−3.43 to −4.86	−4.87 to −6.51
	Second Division	>−0.52	−0.52 to −1.74	−1.75 to −3.01	−3.02 to −4.22	−4.23 to −5.65
	Combinate	>−0.37	−0.37 to −1.13	−1.14 to −2.07	−2.08 to −3.23	−3.24 to −4.77
Impacts (g force)	First Division	<1	1 to 1.99	2 to 2.99	3 to 4.99	5 to 8
	Second Division	<1	1 to 1.99	2 to 2.99	3 to 4.99	5 to 8
	Combinate	<1	1 to 2.99	3 to 4.99	5 to 6.99	7 to 10

**Table 2 sensors-22-00324-t002:** Quality result of the k-means clustering.

Variables		Mean Square	gl	F	Sig.
Speed (km/h)	First Division	634,321.87	4	567,982.49	<0.001
	Second Division	788,099.98	4	848,123.12	<0.001
	Combinate	2,100,865.21	4	2,241,568.09	<0.001
Accelerations (m/s^2^)	First Division	2127.11	4	15,297.90	<0.001
	Second Division	1178.55	4	13,565.70	<0.001
	Combinate	3339.22	4	27,971.25	<0.001
Decelerations (m/s^2^)	First Division	1885.18	4	12,038.13	<0.001
	Second Division	1147.24	4	10,000.80	<0.001
	Combinate	3161.03	4	35,059.21	<0.001
Impacts (g force)	First Division	14,097.12	4	164,510.54	<0.001
	Second Division	10,801.41	4	135,620.06	<0.001
	Combinate	22,701.24	4	92,375.76	<0.001

**Table 3 sensors-22-00324-t003:** Differences between competitive level in external workload according to the volume of demands.

Variables	Intensities	First Division			Second Division			*p*	*Z*	*d*
		Mdn	Range		Mdn	Range				
			Lower	Upper		Lower	Upper			
Duration (seconds) *		472.00	450.00	528.00	370.00	370.00	384.00	**<0.01**	−0.96	0.74
Speed Zones (m)	Total *	484.50	457.00	508.70	324.50	281.80	378.60	**<0.01**	−7.67	0.82
	Standing *	41.30	36.50	43.70	32.20	29.40	34.40	**<0.01**	−4.52	0.46
	Walking *	137.00	129.30	142.30	98.00	92.50	105.60	**<0.01**	−7.25	0.78
	Jogging *	133.40	126.40	143.20	88.20	82.60	97.40	**<0.01**	−7.92	0.86
	Running *	109.80	105.30	115.90	58.10	50.40	82.60	**<0.01**	−7.40	0.80
	Sprinting *	60.60	56.50	64.90	44.70	39.80	58.50	**<0.01**	−2.77	0.28
Accelerations (count)	Total *	214.00	196.00	228.00	160.00	154.00	176.00	**<0.01**	−6.57	0.69
	Very low *	68.00	63.00	73.00	56.00	54.00	60.00	**<0.01**	−5.39	0.56
	Low *	94.00	88.00	102.00	68.00	64.00	75.00	**<0.01**	−6.87	0.72
	Moderate *	40.00	38.00	43.00	26.00	25.00	29.00	**<0.01**	−7.74	0.84
	High *	9.00	8.00	10.00	7.00	6.00	9.00	**<0.01**	−3.66	0.37
	Very high	1.00	1.00	2.00	1.00	1.00	2.00	0.51	−0.67	0.07
Decelerations (count)	Total*	141.00	132.00	152.00	93.00	87.00	100.00	**<0.01**	−9.50	1.07
	Very low *	40.00	38.00	43.00	31.00	28.00	37.00	**<0.01**	−5.45	0.56
	Low *	53.00	49.00	59.00	32.00	30.00	35.00	**<0.01**	−10.09	1.16
	Moderate *	30.00	29.00	32.00	16.00	15.00	18.00	**<0.01**	−10.57	1.24
	High *	13.00	13.00	15.00	8.00	7.00	10.00	**<0.01**	−7.14	0.77
	Very high	4.00	4.00	5.00	4.00	4.00	6.00	0.31	−1.02	0.10
Impacts (count)	Total *	1076.00	1043.00	1143.00	704.00	626.00	820.00	**<0.01**	−8.44	0.93
	Very low *	438.00	420.00	463.00	300.00	257.00	328.00	**<0.01**	−8.41	0.92
	Low *	506.00	478.00	534.00	333.00	291.00	372.00	**<0.01**	−7.88	0.85
	Moderate *	102.00	96.00	110.00	59.00	48.00	72.00	**<0.01**	−8.45	0.93
	High *	29.00	27.00	32.00	16.00	13.00	19.00	**<0.01**	−7.14	0.77
	Very high *	7.00	7.00	8.00	3.00	3.00	5.00	**<0.01**	−6.22	0.68

Note. Mdn: median; *p*: *p*-value; Z: Mann–Whitney U value; d: Cohen’s d effect size; mag: interpretation of Cohen’s d effect size. * Statistical differences between teams. Bold values represent statistical differences between teams.

**Table 4 sensors-22-00324-t004:** Differences between competitive levels in external workload according to the intensity of demands.

Variables	Intensities	First Division			Second Division			*p*	*Z*	*d*
		Mdn	Range		Mdn	Range				
			Lower	Upper		Lower	Upper			
Speed Zones (m/min)	Total	60.24	58.31	61.72	57.96	51.74	65.37	0.09	−1.72	0.17
	Standing	5.11	4.92	5.24	5.23	4.83	5.57	0.40	−0.84	0.08
	Walking	17.15	16.65	17.49	17.43	16.18	18.53	0.92	−0.97	0.10
	Jogging *	16.63	15.57	17.22	15.64	14.69	16.81	**0.02**	−2.19	0.22
	Running *	13.89	12.99	14.40	10.76	9.38	13.19	**<0.01**	−3.75	0.38
	Sprinting	7.28	6.49	7.95	6.98	6.00	10.18	0.89	−0.14	0.01
Accelerations (count/min)	Total *	27.44	26.84	28.10	25.56	24.78	26.67	**0.01**	−2.53	0.26
	Very low	8.55	8.36	8.85	8.89	8.50	9.33	0.12	−1.56	0.15
	Low *	12.24	11.88	12.62	11.05	10.05	11.66	**<0.01**	−3.31	0.34
	Moderate *	4.99	4.80	5.16	4.13	3.89	4.63	**<0.01**	−3.91	0.40
	High	1.16	1.08	1.22	1.09	1.00	1.25	0.92	−0.10	0.01
	Very high	0.11	0.09	0.15	0.15	0.00	0.18	0.51	−0.66	0.06
Decelerations (count/min)	Total *	17.90	17.23	18.48	15.58	14.43	16.64	**<0.01**	−5.49	0.57
	Very low	4.93	4.77	5.18	5.10	4.86	5.74	0.41	−0.82	0.08
	Low *	6.58	6.32	6.89	5.32	4.74	5.56	**<0.01**	−7.21	0.77
	Moderate *	3.62	3.43	3.75	2.55	2.33	2.77	**<0.01**	−7.73	0.84
	High *	1.75	1.68	1.88	1.30	1.19	1.48	**<0.01**	−4.03	0.42
	Very high	0.53	0.46	0.58	0.64	0.61	0.74	0.07	−1.82	0.18
Impacts (count/min)	Total *	134.47	129.89	138.58	120.89	113.62	138.81	**<0.01**	−2.84	0.30
	Very low *	55.17	53.14	56.89	50.00	45.19	56.30	**0.01**	−2.68	0.27
	Low *	60.74	58.49	64.63	57.45	51.06	63.24	**0.01**	−2.54	0.26
	Moderate *	12.28	11.92	12.76	10.08	8.94	11.30	**<0.01**	−4.50	0.46
	High *	3.40	3.08	3.60	2.67	2.04	3.08	**<0.01**	−4.01	0.42
	Very high *	0.76	0.68	0.88	0.47	0.37	0.63	**<0.01**	−4.18	0.43

Note. Mdn: median; *p*: *p*-value; Z: Mann–Whitney U value; d: Cohen’s d effect size; mag: interpretation of Cohen’s d effect size. * Statistical differences between teams. Bold values represent statistical differences between teams.

## Data Availability

The data presented in this study are available on request from the corresponding author. The data are not publicly available due to the Organic Law 3/2018, of 5 December, on the Protection of Personal Data and Guarantee of Digital Rights of the Government of Spain, which requires that this information must be in custody.

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
