# Peer review of "Individualization of Intensity Thresholds on External Workload Demands in Women’s Basketball by K-Means Clustering: Differences Based on the Competitive Level"

_sensors, 2022, doi:10.3390/s22010324_

Round 1

Reviewer 1 Report

This study has the following aims (a) to realize k-means cluster algorithm in two professional female basketball teams (first and second division) to know the different training intensities in speed, acceleration, deceleration, and impacts; (b) to characterize the external workload demands of the selected variables in total and normalized variables, and (c) to analyze the differences between two teams on the time considering the load supported in the 5v5 game situations in the training sessions. The strengths of this article lie in the fact that it categorizes them as performances based on data collected from the sample itself in the context of training, and not only based on previous studies. In addition, they provide information on the workload that the players supported in the face of the same stimulus (training game) in which the demands are different depending on the competitive level. Below, I suggest some points to the authors to improve the article.

- In general, the authors make a good introduction, contextualize the game, and discuss previous studies that sought to categorize acceleration, deceleration, velocity, and impact zones in formal games. The lacking point is the justification for the evaluation of these variables in the training context, considering that the cited articles prioritized the study of these variables in formal games.

- I suggest changing the description of the objectives in the introduction according to the abstract, as I believe the objectives are better described in the abstract.

- In the description of the study design, there is no mention of the number of sessions in which the assessment was carried out.

- When the authors describe the participants, they divide the groups into first and second divisions, however throughout the article the terms “elite” and “professional” are used to distinguish the investigated groups. I suggest standardizing the name of the groups.

- In line 159, it is said that the 5vs5 games evaluated took place on the whole court, however, in figures 1 and 2, there is information that athletes were evaluated in small-sided games. I suggest reviewing it.

- In the methodology, nothing is informed about how long the 5vs5 games lasted, and this information only appears at the conclusion (lines 279 and 280). Another information presented lately, in the discussion, is the fact that there were no breaks in the 5vs5 game (line 239). I suggest reviewing these points.

- in line 174, there is a description of the colors of the information in the histograms, however, this information is unnecessary in the methodology, as it is already explicit in the legend of the figure.

- in lines 185 and 186 it is said that the division of groups for the elaboration of table 1 was based on previous studies, however, they are not indicated there. Also, why is this information highlighted in italics in the text?

- In table 1, I suggest the authors take care with the intervals that make up each table cell. For example: if an athlete performed an acceleration of 2.01 m/s², would it be categorized as “walking” or “jogging”? Note that the same value is in both columns, finishing and starting, respectively, the data interval, and this happens in a lot of cases. I suggest reviewing it.

- In Table 2, the effect size of the intensity of acceleration, deceleration, and impact zones was not large; what would be the justification? Could it be, for example, related to being a training situation and not an official game? This aspect was not discussed in the article.

- In general, I missed the discussion of the main findings of the study, as well as the absence of a more direct response to the objectives of the study, discussing them based on the data obtained. The discussion is quite general, and practically takes up the framework of the study.

- I think the information presented in the conclusion could be moved to the discussion, leaving the conclusion session with a summary of the study's findings, as well as pointing out the limitations of the study, such as the non-evaluation of the occurrence of these actions in official games of both groups, and the possibility of evaluating these variables in other periods of the training periodization in future studies.

- In point 1 of the conclusion, only general performance data are highlighted. I suggest comparing the performance of both groups and discussing it.

- In point 2 of the conclusion, between lines 275 and 276, there is information that elite athletes had higher thresholds for all types of movement and all intensities. However, table 2 shows professional athletes’ higher thresholds ​​in some cases, for example in standing and walking. I suggest reviewing the writing of the statement.

Author Response

Consulte el archivo adjunto.

Reviewer 2 Report

The manuscript is devoted to the application of the k-means clustering algorithm for individualization of intensity thresholds on external workload demands in female basketball. The topic is interesting and actual. The abstract briefly reflects the paper content. The Introduction, to my mind, is written correctly. The structure of the manuscript corresponds to this type of publication. However, to my opinion, the manuscript should be improved before acceptance. Below, I present my remarks.

  1. The English grammar and style should be improved. There are many mistakes. For example, Title:" based on competitive level". Should be: " based on a/the competitive level". Row 15-16: “impact (IMP) zones have been almost created according to arbitrary thresholds, not considering the specific workload profile”: What is mean “almost?” The zones can be created or not created. Then, it will be better to change “not” to “without”, etc in the whole manuscript.
  2. The section "Materials and Methods" should be reconsidered. This section should contain in terms of my view the structure block chart of the stepwise procedure that is implemented within the framework of the research, the mathematical foundation and equations including clustering quality criteria, describing the statistical techniques etc.
  3. It is necessary to add the section "Experiment", were to present in detail the description of the experiment carried out.
  4. And the main question. Why k-means algorithm? Did you try other clustering algorithms? Where are the results regarding the evaluation of the clustering quality using appropriate criteria? This information is absent. Thus, to my mind, the section results should be rewritten too.

Round 2

Reviewer 1 Report

Congratulations to the authors for this new version of the manuscript. All my notes and suggestions were satisfactorily corrected or commented on in the cover letter.

Reviewer 2 Report

Thanks, I have no other questions